# Ground Deformations Controlled by Hidden Faults: Multi-Frequency and Multitemporal InSAR Techniques for Urban Hazard Monitoring

**Federica Murgia [1],\*, Christian Bignami [2] , Carlo Alberto Brunori [2] , Cristiano Tolomei [2] and Luca Pizzimenti [2]**

1   Dipartimento di Ingegneria dell'Informazione, Elettronica e Telecomunicazioni (DIET), La Sapienza University of Rome, 00184 Rome, Italy
2   Istituto Nazionale di Geofisica e Vulcanologia, 00143 Roma, Italy; christian.bignami@ingv.it (C.B.); carloalberto.brunori@ingv.it (C.A.B.); cristiano.tolomei@ingv.it (C.T.); luca.pizzimenti@ingv.it (L.P.)
\*   Correspondence: federica.murgia@uniroma1.it

**Abstract:** This work focuses on the study of land subsidence processes by means of multi-temporal and multi-frequency InSAR techniques. Specifically, we retrieve the long-term evolution (2003–2018) of the creeping phenomenon producing ground fissuring in the Ciudad Guzmán (Jalisco state, Mexico) urban area. The city is located on the northern side of the Volcan de Colima area, one of the most active Mexican volcanoes. On September 21 2012, Ciudad Guzmán was struck by ground fissures of about 1.5 km of length, causing the deformation of the roads and the propagation of fissures in adjacent buildings. The field surveys showed that fissures follow the escarpments produced during the central Mexico September 19 1985 Mw 8.1 earthquake. We extended the SAR (Synthetic Aperture Radar) interferometric monitoring starting with the multi-temporal analysis of ENVISAT and COSMO-SkyMed datasets, allowing the monitoring of the observed subsidence phenomena affecting the Mexican city. We processed a new stack of Sentinel-1 TOPSAR acquisition mode images along both descending and ascending paths and spanning the 2016–2018 temporal period. The resulting long-term trend observed by satellites, together with data from volcanic bulletin and in situ surveys, seems to suggest that the subsidence is due to the exploitation of the aquifers and that the spatial arrangement of ground deformation is controlled by the position of buried faults.

**Keywords:** subsidence; multi-temporal analysis; PS; SBAS; InSAR; urban monitoring; buried faults

## 1. Introduction

Ground subsidence is a geological phenomenon occurring in both uninhabited [1,2] and densely populated regions [3], or coastal plains [4,5]. Often, the subsidence of ground surfaces can be the result of the natural compaction of sediments or caused by anthropological activities like the extraction of groundwater, geothermal fluids, oil, gas, coal and other solids through mining [6]. Even if the hazard associated with subsidence is often different from the ones related to sudden events like earthquakes [7] because of their influence over a wide area, or like sinkholes formation, because of the limited extension of the affected area [8], the damages connected to surface slow sinking events are, however, extensive and with impacts across wide regions. Ciudad Guzmán (CG) is located in the Colima Volcanic Complex, near the active volcano Fuego de Colima [9] (Figure 1). It is a densely populated city characterized by moderate seismicity [10,11] and susceptible to intense withdrawal of water from aquifers widespread for civil and industrial purposes [12]. The already observed subsidence can be controlled by the presence of buried faults that can guide the generation of the

fissuring in the urban area [13]. The latter phenomenon produces a significant hazard that needs to be accurately assessed and monitored in order to prevent future impact on civil infrastructures. In order to analyze such surface movements, SAR Interferometry (InSAR) has proved to be a particularly useful tool capable of mapping ground deformation with a very high spatial resolution, with high accuracy [7,8,13]. In this study, in the framework of the time-series analysis of ground deformation, we applied two different interferometric techniques, i.e., Permanent Scatterers (PS), and Small BAseline Subset (SBAS) to the SAR dataset acquired by the European Space Agency (ESA) mission Sentinel-1. With this new set of ground deformation measurements, we extend the InSAR analysis of the CG ground deformations observed between 2003 and 2016 and described in [13,14], focusing on the most recent 2016–2018 period.

The final aims of this paper are to determine the correlation between the shapes of subsidence deforming and fracturing in Ciudad Guzmàn town with the position of the faults dislocating the bedrock under unconsolidated deposits filling the valley; to describe the ground subsidence evolution in urban areas by means the integration of SAR data acquired by different satellites platforms and sensors (Envisat, COSMO SkyMed, and Sentinel-1—in ENV, CSK and S1, respectively) and processed with multi-temporal InSAR techniques; to define a methodological approach for the study and the prevention of these hazardous geological processes.

## 2. Geological Overview

Ciudad Guzmán (1.500 m above sea level), with a population of about 105.000 inhabitants, is a Mexican city belonging to the Zapotlàn El Grande municipality in the Jalisco state. The town is located in the eastern side of a tectonic valley, the so-called Colima Graben [9]. The Colima Graben is the southern branch of the Colima-Tepic-Chapala triple junction located in the western sector of the Trans-Mexican Volcanic Belt (TMVB). The TMVB is a 1200 km long active continental volcanic arc originated by the subduction of the Cocos and Rivera plates along the Middle American Trench [15]. The TMVB is structurally divided into several regions, one of which is the Colima Graben (or Colima Rift). The Colima Graben is a structure that consists of three segments, the Northern Colima Graben (NCG), the Central Colima Graben (CCG) and the Southern Colima Graben (SCG). This region is considered the eastern limit of the Jalisco Block. The distinctive tectonic feature of this area is the presence of three structural patterns with NW, SW and EW orientation, associated with the local recent rifting processes [16,17].

Ciudad Guzmán (CG) is located in the South-Eastern part of the NCG and is surrounded by reliefs consisting of Late Miocene-Pliocene volcanic deposits, Jurassic-Eocene sedimentary and intrusive rocks [9]. The valley is filled by a sequence of quaternary lacustrine sediments, alluvium, colluvium and volcanic deposits of the nearby Colima Volcanic Complex (CVC), one of the most active volcanoes of the TMVB. The NCG (60 km long and 20 km wide) is flanked by parallel NNE-SSW-trending active faults dipping 70° toward the graben axis and is mainly characterized by a normal component of motion with respect to lateral strike-slip one.

The inhabited area of CG, as shown in Figure 1, is bound to the North by the wetland of Zapotlán basin and the sequence of sediments under CG urban area is mainly composed of weakly lithified volcaniclastic deposits characterized by a relatively low cohesion [10] as reported in Figure 2. This area is exposed to hazardous natural events such as landslides, volcanic eruptions and earthquakes [11], and it was hit by several seismic events in the past years (1911, 1931, 1932, 1941, 1975, 1985 and 2003) [18], mainly due to the tectonic activity of the volcanic arc and subduction zone [15–17]. In particular, the 1985 Mexico City earthquake (Mw 8.1) generated ground fissuring in the urban area of CG, even though the epicenter was located in Michoacán State, about 190 km far from CG. On 21 September 2012, without any correlation with seismic events, an alignment of soil fractures was formed in the city of CG. The orientation and position of the 2012 fractures are compatible with the ones opened during the 1985 earthquake.

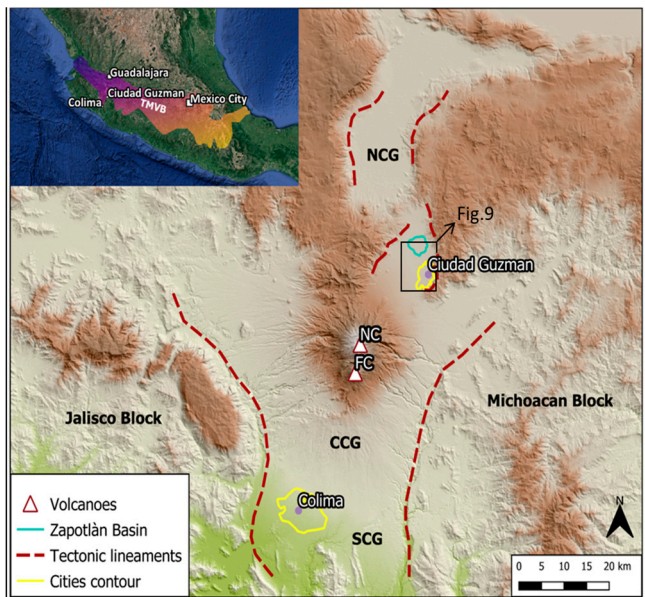

**Figure 1.** The location of the study area. The upper left Google Earth view localizes the Colima and Ciudad Guzmàn regions (in the Colima Graben area) with respect to the Middle America trench (Central-Western Mexico). The physical map shows the subdivision of the Colima Graben (CG) in the three segments, namely, the Northern Colima Graben (NCG), Central Colima Graben (CCG) and Southern Colima Graben (SCG) between the Jalisco and the Michoacán blocks. Ciudad Guzmàn is located in the South-Eastern part of the NCG, and in the Northern side of the Nevado de Colima (NC) and Fuego de Colima (FC) volcanic structures (the Colima Volcanic Complex-CVC).

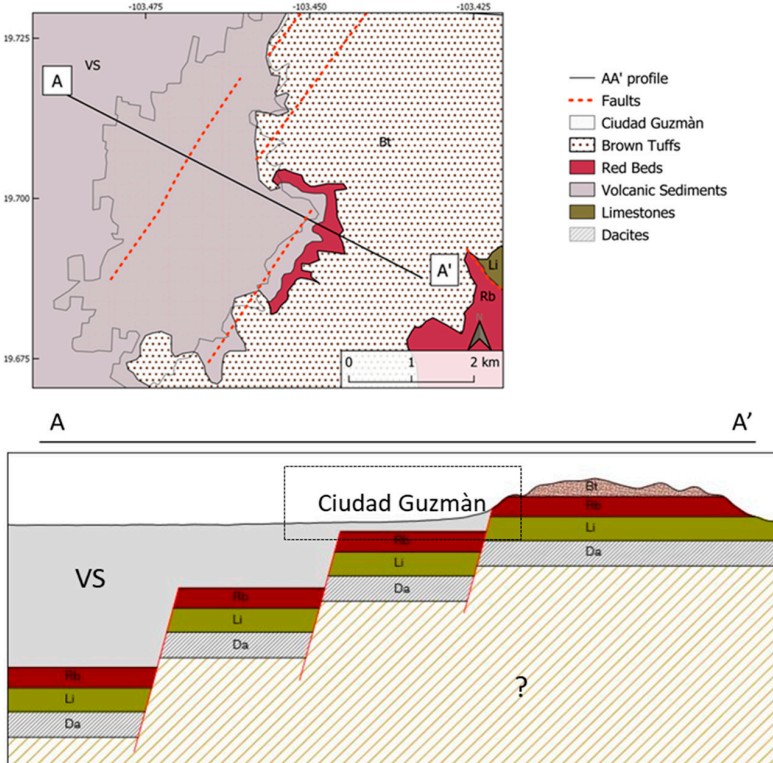

**Figure 2.** Simplified Geological map of Ciudad Guzman. (VS) Volcanic Sediments; (BT) Brown Tuffs—Plio-Quaternary deposits; (Rb) Red beds—Late Cretaceous; (Li) Limestone; (Da) Dacites—Early Cretaceous. On the bottom, the schematic geological cross-section relative to the AA' profile in Figure 2a (modified from [19]).

## 3. Materials and Methods

We processed two S1 SAR stacks of 53 and 70 S1 images (TOPSAR mode) on ascending and descending paths (the orbit numbers are 49 and 12, respectively) covering a temporal span from November 2016 to April 2018. Two different multi-temporal techniques, both implemented in SARScape© software (Sarmap SA©), were adopted for the datasets: PS [20–22] and SBAS [23–25]. As far as the SBAS method, all the co-registered interferograms were obtained by applying a multi look factor of 4 × 1 in range and azimuth directions, leading to a 15 × 15 m pixel size on the ground. Then, the interferograms were computed and only the pixels showing a coherence value greater than 0.35 were unwrapped for each pair.

We also considered a stack of 98 CSK images spanning a time interval from October 2011 to September 2015 [14,26], which were processed with the IPTA multi-baseline technique [27] implemented in GAMMA© Software to integrate the temporal analysis between ENV and S1 (Figure 3). The above-mentioned method computes a stack of interferograms, which are generated considering only the SAR pairs characterized by values of spatial and temporal baselines limited within specific ranges, aiming at minimizing the interferometric coherence loss. Then, the deformation time-series and residual topographic heights are estimated by using the Singular Value Decomposition (SVD) Least-Squares inversion technique [25] applied to the interferometric stack [13].

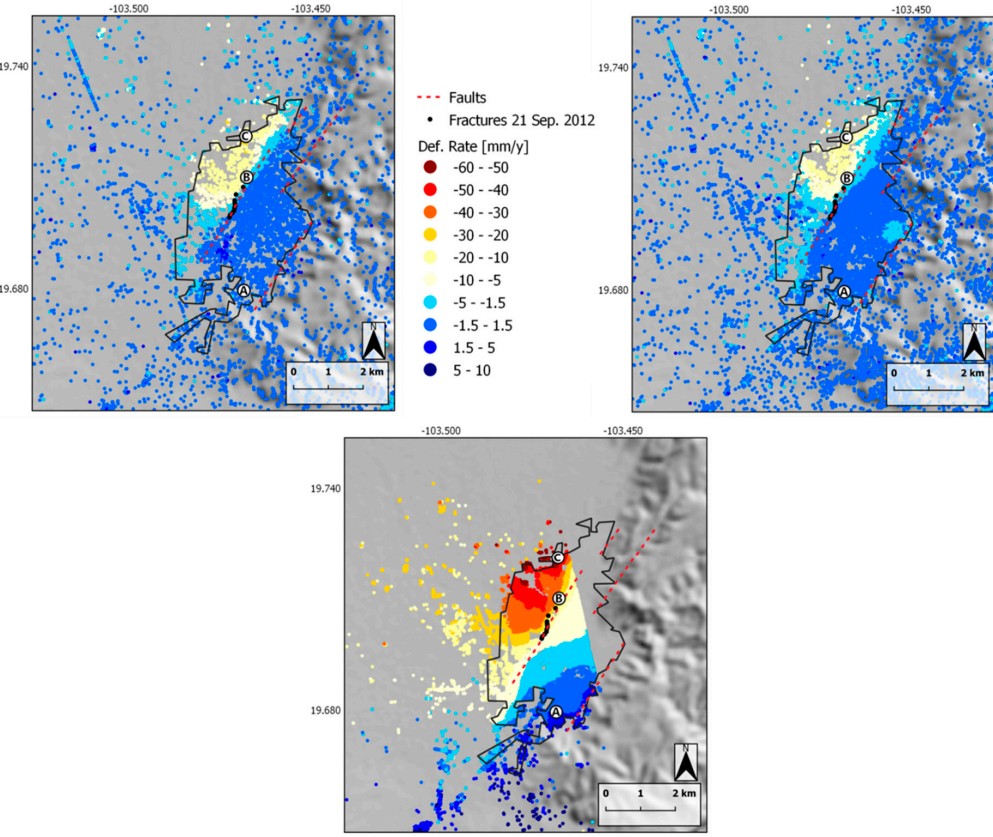

**Figure 3.** Maps of the deformation rate from the ENV dataset (2003–2010 time period descending and ascending paths, upper panels on the left and right, respectively) and from the CSK dataset (2011–2015 time period ascending path, lower panel). Both the ENV and CSK datasets were processed using the IPTA multi-baseline technique. The white circles indicate three selected sites individuating different displacements rates: (A) stable or light-positive ground velocities (toward the satellite); (B) intermediate values of negative velocities (moving away from the satellite), near the fractured area; (C) highest values of negative ground velocities. A ground urban survey performed in CG in November 2012 assessed the deformations and associated fractures (black points alignment on the maps) that occurred on 21 September 2012 [13].

### 3.1. Sentinel-1 SBAS Processing

In order to track the November 2016–April 2018 temporal evolution of ground subsidence affecting CG, we applied the Small BAseline Subset (SBAS, [23–25]–Figure 4a) technique to process the time series of S1 dataset. The SBAS algorithm is based on an appropriate combination of differential interferograms produced by data pairs characterized by a small spatial (perpendicular) baseline and limited temporal separation in order to limit the spatio-temporal decorrelation phenomena. At the beginning of the SAR data processing, we selected a maximum spatial baseline equal to 150 m and a maximum temporal baseline equal to 72 days and resulting in 160 ascending and 208 descending pairs. The SBAS method computes deformation time series and residual topographic heights using the Singular Value Decomposition (SVD) Least-Squares inversion technique [23,24]. It is worth saying that, in order to obtain a cleaner final displacement map, after the first time-series estimation, we applied a custom atmospheric filtering to the stack of unwrapped interferograms. The dimensions of the used filter are 1200 m × 365 days, representative of a low pass filter in space and a high pass filter in time.

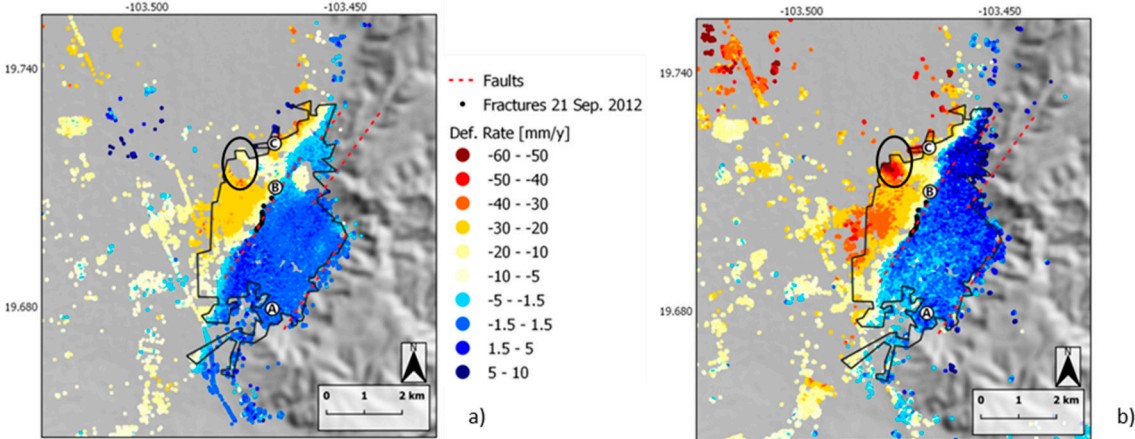

**Figure 4.** Deformation rate maps from the 2016–2018 S1 dataset (ascending path). PS (**a**) and SBAS (**b**) techniques results. The white circles indicate the selected sites individuating different displacements rates: (A) stable; (B) intermediate/fractured; (C) highest values. The black ellipses surround a particular area of CG that will be discussed in the Discussion section.

As already mentioned, the above described multi-temporal processing was done with the SBAS technique implemented into the Sarscape© software. As it is well known, the phase unwrapping step, foreseen only in the SBAS technique and not in the PS one, reports the 2π module interferograms to a continuous phase representation, according to the minimization of a proper cost function. The algorithm starts to unwrap the pairs from a point. The so-called reference point is assumed stable during the period of interest. This implementation does not permit to manually select the reference, but applies a re-flatting of the interferometric stack, based on the selection of some control points in the unwrapped dataset. This step can lead to unaccounted bias in the final products, if the re-flattening did not work fine. To overcome this issue, we scaled the deformation rate map and the time series of deformation according to the part of the city (i.e., the south-eastern part of CG) that showed a stable behavior in the whole dataset, including the PS one. More precisely, we selected a group of points contained in a circular area of 250 m diameter, characterized by the same characteristics in terms of topography, and a very similar deformation trend, and we scaled the entire product by the average of the deformation rates and trends retrieved in that area.

### 3.2. Sentinel-1 PS Processing

We also processed the same S1 dataset with the multi-temporal methodology, namely, the Persistent Scatterers (PS [20–22]–Figure 4b) implemented in SARScape© software. PS is an opportunistic

deformation measurement method, i.e., it is able to measure deformation only over the available PSs. Usually, the PS density is low in vegetated, forested and low-reflectivity areas (e.g., very smooth surfaces), and in steep terrain facing the radar sensors [20]. By contrast, PSs are usually abundant on buildings, or urban areas in general, so this multi-temporal technique represents a suitable tool for our study. As it will be discussed later, the PS algorithm implemented in the above mentioned software does not perform the unwrapping, meaning that it algorithm avoids to analyze the phase, since at this stage it may contain unknown signal contributions. Therefore, the degree of "reliability" of each single pixel is evaluated looking at the amplitude dispersion index (i.e., the ratio between the temporal standard deviation of the amplitude and the temporal average of the amplitude), meaning that a pixel characterized by a similar amplitude during all acquisitions is expected to have a small phase dispersion [28].

As usual, in order to reduce the atmospheric disturbances, a spatial and temporal filter is applied. The parameters we selected are the same set for SBAS elaboration.

## 4. Results

As already mentioned, ground deformation time series of the S1 dataset were evaluated by means of both the SBAS and PS techniques. The obtained LOS (Line of Sight) mean ground velocity maps (Figure 3) highlight the presence of a subsidence (i.e., the distance satellite-target increases) in the NW area of CG, in agreement with the results shown in [13]. This subsiding area is bordered to the East by the location of the 2012 surface fissures. Crossing the rupture alignment, we found a stable area (South-Eastern side of CG), as observed in [13] with ENV and CSK deformation time series (Figure 2). As in [13], we considered three sites, (A; B and C, Figure 3) to analyze the ground deformation behavior in different parts of the city. Specifically, point (A) is located in the SE part of GC, where the displacements range from 0 to slightly positive values in the satellite LOS (i.e., distance satellite-target reduced); site (B) is located near the Northern 2012 fissure alignment observed during the field survey of November 2012 [13], and site (C) is in the NW part of the city, where the subsidence phenomenon is more pronounced. As it can be noted, even though the two deformation rate maps retrieve the same spatial arrangement in terms of deformation patterns, the values are not equal, especially in the high-rate and less coherent part of the city (i.e., the north-western one) due to the fact that the two techniques are different.

### 4.1. Comparison between SBAS and PS Processing

With the aim of cross-validating the results achieved using different technique applied to the same area and the same SAR dataset, we compare the results of PS and SBAS techniques. The PS and SBAS retrieved velocities patterns are very similar, although they show some discrepancies in terms of deformation rate values mainly due to the two different implementations. This implies that the procedure of selecting the coherent points and in particular the connection graph of the pairs to evaluate are not the same, possibly leading to a bias in the deformation rates values. Actually, SBAS keeps in account discrete targets and the backscattered response to the radar echo is formed by the signal backscattered from all the targets present into the resolution cell. Instead, the PS methodology considers only the response over the time interval for each single scatterer, also at a subpixel scale. The difference in terms of spatial coverage is due to the fact that SBAS, as already stated, keeps in account for distributed scatterers. Indeed, the interferograms are multi-looked and spatially filtered with Goldstein noise reduction filter (that is a mandatory step for this kind of multi-temporal processing); therefore, the coherence increases in SBAS, and consequently the spatial coverage increases as well. This is particularly true in the area where fast subsidence occurs: over these regions, the PS method, which does not unwrap the interferograms, is unable to "follow" such trend, thus it loses possible persistent scatterers candidates. In particular, the retrieved PS maps show a deformation rate up to 15 mm/year in correspondence of the fractures, and a faster subsidence reaching 40 mm/year in the NW of CG where the velocity retrieved from the SBAS processing reaches 45 mm/year. The SBAS and PS

results were compared (Figure 5). In the scatterplots of Figure 6 are reported the velocities of all the scatterers included in a circular area of 250 m of radius surrounding each of the three considered sites (named A, B and C) and highlighting three different displacement rates.

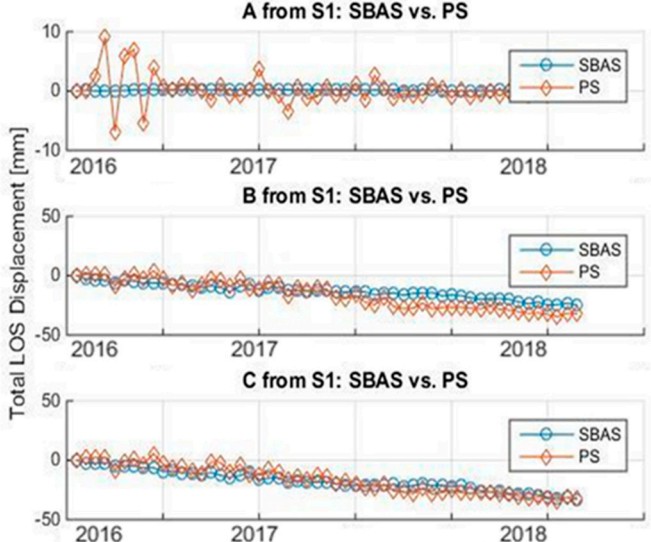

**Figure 5.** Mean LOS ground velocity of S1 ascending stack: the SBAS vs. PS cross-comparison analysis for the A, B and C selected sites of Figure 4.

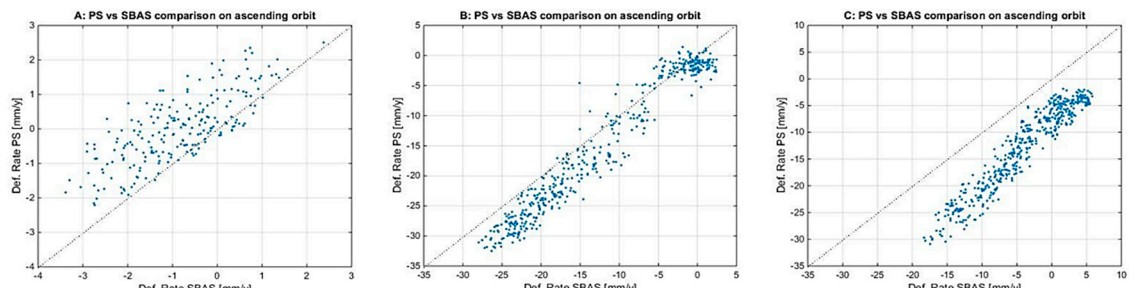

**Figure 6.** LOS ground velocity of the S1 ascending stack: the SBAS vs. PS cross-comparison analysis for the A, B and C selected sites of Figure 4.

The scatterplots highlight different biases and dispersions depending on the analyzed area characterized by different deformation rates.

In Table 1 statistics relative to the comparison of the two multi-temporal techniques adopted in this work for the three scatterplots of Figure 6 are summarized.

The deformation rates of the A site show a high standard deviation difference if compared with their dynamic. This is because the SBAS approach provides a smoother solution with respect to the PS one, and this is more evident in a stable area. The reason may be found in the fact that the SBAS algorithm applies the Goldstein filter [29] to multi-looked pairs, thus generating smoother interferograms. Even though the subsidence pattern is nearly identical for both methods, in the area where the subsidence phenomenon is relevant, the PS method seems to overestimate the deformation rate compared with the SBAS result. However, the maximum deviation between the two multi-temporal techniques is less than 9 mm/yr.

**Table 1.** Cross validation of the two multi-temporal outputs used for the S1 data processing methods. The values are relative to the total component of the deformation rate in the LOS of S1 on the ascending path. The same comparison could not be made for the descending path because of the lack of CSK images on that orbit.

| # | Bias [mm/y] | Std Difference [mm/y] | Correlation |
|---|---|---|---|
| A | 0.832 | 0.731 | 0.767 |
| B | 2.903 | 2.794 | 0.975 |
| C | 8.866 | 2.587 | 0.957 |

*4.2. Deformation Time-Series Analysis*

Aiming at completing the multi-temporal InSAR study carried out by means of the ENV [13] and CSK [14,26] datasets, we added the information retrieved by exploiting the S1 acquisitions along the ascending path only, processed with the SBAS algorithm. For this aim, we first had to fill the gaps in between the three time-series obtained from the different sensors. The problem here was to find the way to link the series, since the three datasets are time-gapped [30]. For the sake of clarity, it is worth saying that the analysis described hereafter refers to the three control points (i.e., A, B, and C) identified and presented above, but the procedure can be applied to the whole map if there is a consistent number of high coherence pixels in common between the different time-series. We first assumed a linear trend for the deformation (Figure 4) since the dynamic of the subsidence resembles a linear evolution with the time. In this way, the gaps between each time series (ENV, CSK, and S1) were filled-in through a linear fitting and by extrapolating the deformation slope. In particular, we performed a linear forward prediction of the previous time series up to the time of the first acquisition of the following dataset (e.g., the ground displacement of the ENV time-series has been prolonged until the first acquisition time of the CSK one). Then, the following time-series were scaled to a value given by the corresponding one assumed by the extrapolation of the previous deformation series in correspondence with the time of the first acquisition date of the considered dataset. The scaling step is fundamental since each time series was evaluated independently and so each long-term ground subsidence observations starts from 0 mm displacement.

Clearly, it is not a rigorous procedure, but the lack of ground truth information and of a geological/hydrological dynamic model did not permit us to apply another approach. However, the obtained results, including the changes in the slope of the displacement, are in agreement with the annual reports of the hydrological situation in the administrative region to which CG belongs [31] and with the agreement for the sustainability of the area [32] that will be presented in the Discussion section. It is worth noticing that in order to make the results comparable, we re-projected the S1 and ENV data on the CSK ascending LOS (LOS$_{CSK}$). This implies the assumption that the deformation is only vertical, neglecting the east-west and the north-south components of the deformation [13,14]. Actually, due to the lack of CSK data along the descending path CSK, the East-West component is not retrievable for the overall monitoring period (2003–2018). We can also add that as regards the North-South direction, it is well known that quasi-polar orbit of space-borne SAR has a very weak sensitivity with respect to deformation. The procedure applied to the ENV and S1 time-series refers to the geometry sketched in Figure 7, where the situations of the CSK and S1 satellite are reported.

It is worth noting that the same procedure is applicable to the ENV dataset considering its standard incidence angle (i.e., 23°). Looking at the acquisition geometry, we can see that:

$$S = X - D = C cos\delta - D, \tag{1}$$

Being $\delta$ the angle of the difference of the incidence angles of S1 and CSK satellites. Moreover,

$$D = y cos\delta = L tg\varphi cos\delta = L tg\delta cos\delta, \tag{2}$$

and since $L = Ctg\theta$, the resulting value for $S$ is given by the following relationship:

$$S = Ccos\delta(1 - tg\theta tg\delta),$$ (3)

Figure 8 shows the results for the three considered sites. As it can be easily noted, the retrieved trend from S1 data is very similar to the ones observed from the ENV and CSK datasets. It is worth nothing that subsidence in the last two years seems to drop down.

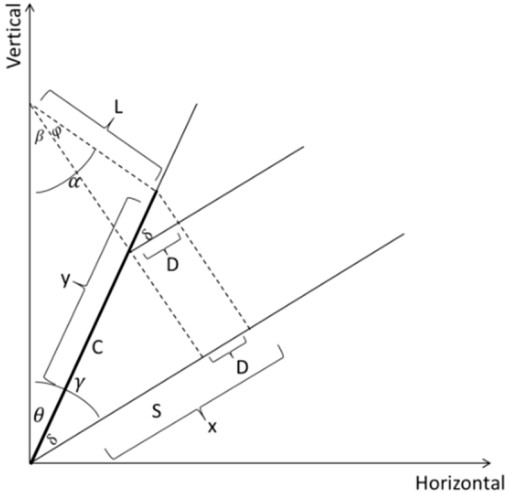

**Figure 7.** Acquisition geometry of the S1 and CSK satellites used to project the vertical component of deformation of the S1 dataset onto the CSK line of sight. It is worth remarking that even though the ENV LOS is not present in this figure, the same approach has been applied to such dataset considering the proper incidence angle, i.e., equal to the standard acquisition off-nadir of 23°.

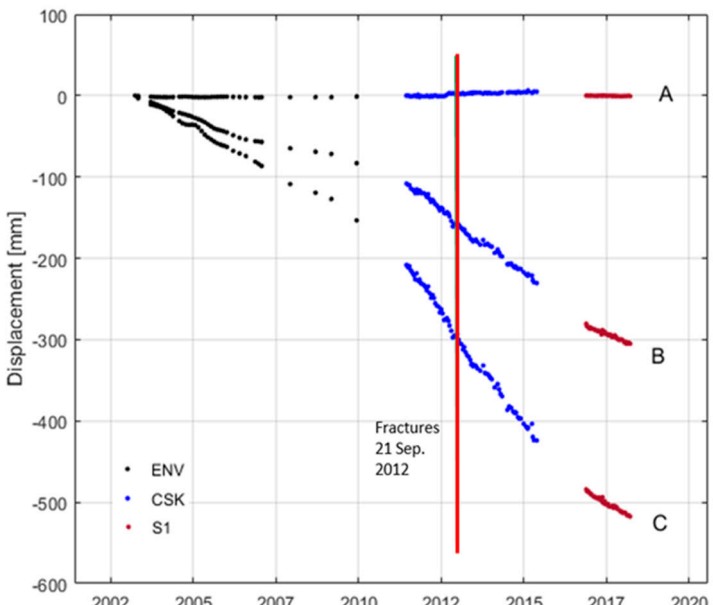

**Figure 8.** Example of three deformation trends in the (**A**), (**B**) and (**C**) sites of Figure 4. The displacements were estimated by ENV, CSK and S1 datasets (from ascending orbit only). The ENV and S1 displacement data are projected along the $LOS_{CSK}$. Each of the three colors (black, blue and red) represents the three satellite-derived displacement measures. The A displacement points are relative to the part of the CG that lies on the bedrock and thus shows a stable behavior. The B displacement points are relative to a site located in correspondence of the cracks that opened on September 2012 (indicated by the green line). The C displacement points are relative to the northwestern part of CG, with high deformation rates.

*4.3. Vertical and East-West Components Estimation*

The vertical (Figure 9) and East-West (Figure 10) deformation rate components were calculated combining the ascending and descending S1 data following the well-known relationships [33] as follows:

$$v_{UP} = \frac{d_{asc}\sin\theta_{desc} - d_{desc}\sin\theta_{asc}}{\sin(\theta_{desc} + \theta_{asc})}$$
$$v_{EAST-WEST} = \frac{d_{desc}\sin\theta_{asc} - d_{asc}\sin\theta_{desc}}{\sin(\theta_{desc} + \theta_{asc})} \tag{4}$$

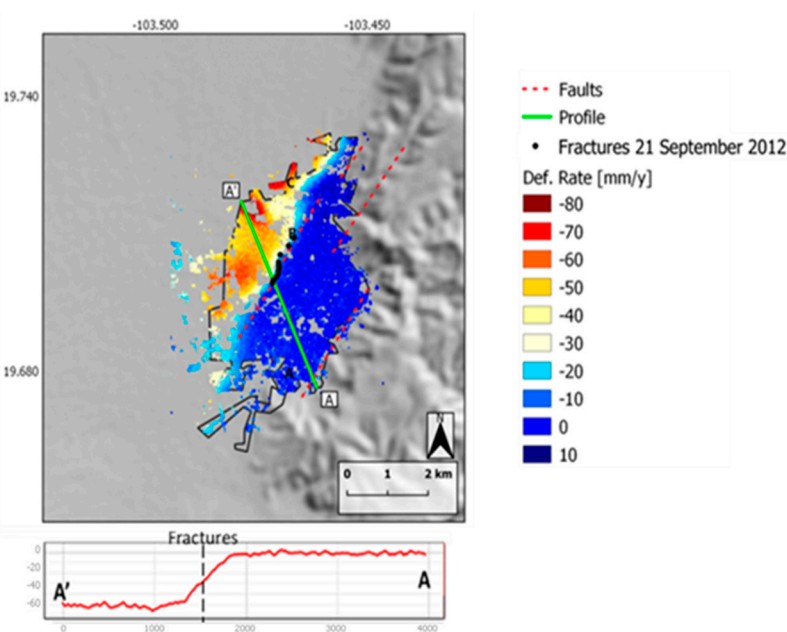

**Figure 9.** Vertical deformation rate retrieved from S1 ascending and descending data. In the A-A' velocity profile, the horizontal axis gives the distances (m), the vertical axis the velocities (mm/yr): positive and negative values indicate Eastward and Westward movements respectively.

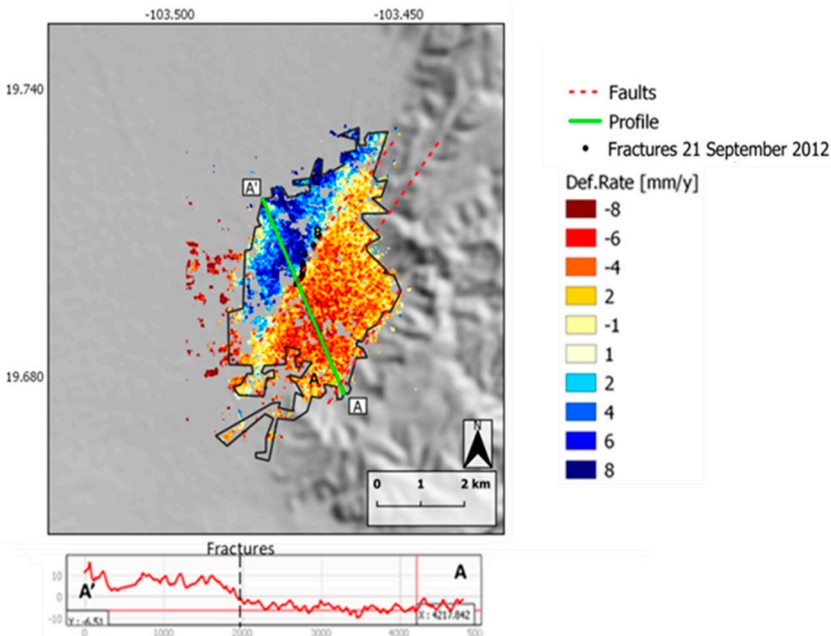

**Figure 10.** East-West deformation rate retrieved from S1 ascending and descending data. In the A-A' velocity profile, the horizontal axis gives the distances (m), the vertical axis the velocities (mm/yr): the negative velocity values identify the subsidence area.

As expected, the movement on the North-Western side of CG is mainly vertical with a small horizontal component of deformation. The maximum vertical deformation rate reached on the velocity profile A-A' is about −60 mm/year, while on the same profile, the E–W deformation rate reached up to 8 mm/year towards the West in the CG Eastern districts and more than 10 mm/year towards the East in the Western districts. As shown by the velocity profiles for both the E–W and the vertical velocity maps, the deformation rate increases in the NW area of CG starting from the fissures, but the Eastern side of the city is almost stable, confirming the rates observed with the previously retrieved surface displacement time series.

## 5. Discussion

Our results confirm that the subsidence, observed between 2003 and 2016 [13,14,26], is still ongoing. The ground fissuring phenomenon seems to maintain the same pattern previously observed [13], showing a considerable extent over a large area located on the northwestern side of CG. The deformation time-series gathered by S1 data do not show a relevant seasonal oscillation related to dry and rainy seasons, so it is unlikely that this could be considered as a possible cause of ground deformations.

As reported in Figure 11, the valley that bounds CG has experienced a relevant increase in the use of the soil for agricultural purposes: the surface dedicated to greenhouse cultivation has doubled in the last 6 years, meaning that the consequent water withdrawal has increased. As reported in [32], under a coordination agreement for the recovery and sustainability of the Lerma-Chapala basin, to which CG belongs, in the last years the situation concerning the aquifer availability is starting to be critical. Indeed, the availability of granted water from the aquifers located in the municipality of Zapotlán el Grande has decreased in the last few years [31,34], justifying the small increase in slope of the time series of the deformation rate.

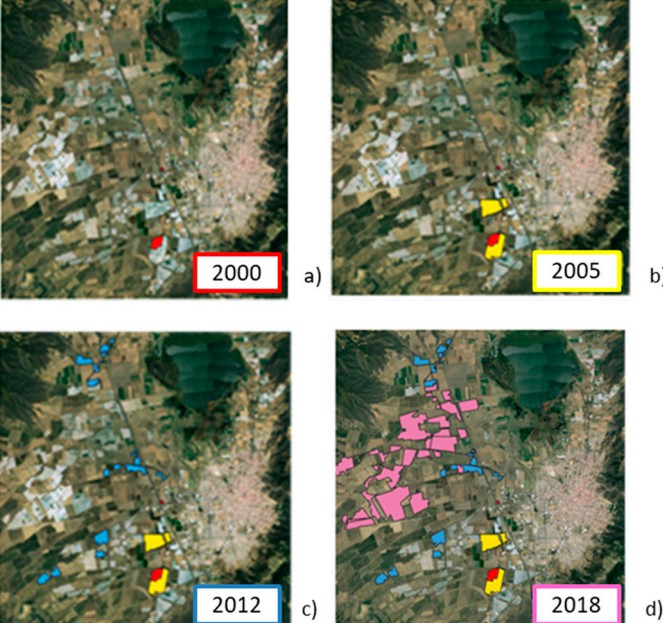

**Figure 11.** Greenhouse "time-lapse" of evolution in the 2000–2018 period (**a**–**d**). Time series of images with the Google Earth Time Slider tool provided by Google Earth Pro (v. 7.1.5.1557).

The demographic trend can play, on the other side, an important role in such a context. Actually, we observed an increase in the population in the 1995–2015 period [31]. As a consequence, the city has been subject to an intense urbanization process in the last few years that has led to a relevant increase in terms of extension. This fact alone cannot justify the subsidence phenomenon since the overall city was subject to the homogeneous urban sprawl, even though it indirectly can reflect on an increase in

water resources exploitation. We observed that the urban sprawl is still ongoing, especially in the part of CG that is experiencing the highest rates of deformation (i.e., the black ellipses sketched in Figure 4, as shown in Figure 12 referring to the 2005–2018 time period. It is worth noting that the considered interval comprises the year of the fractures opening (i.e., 2012).

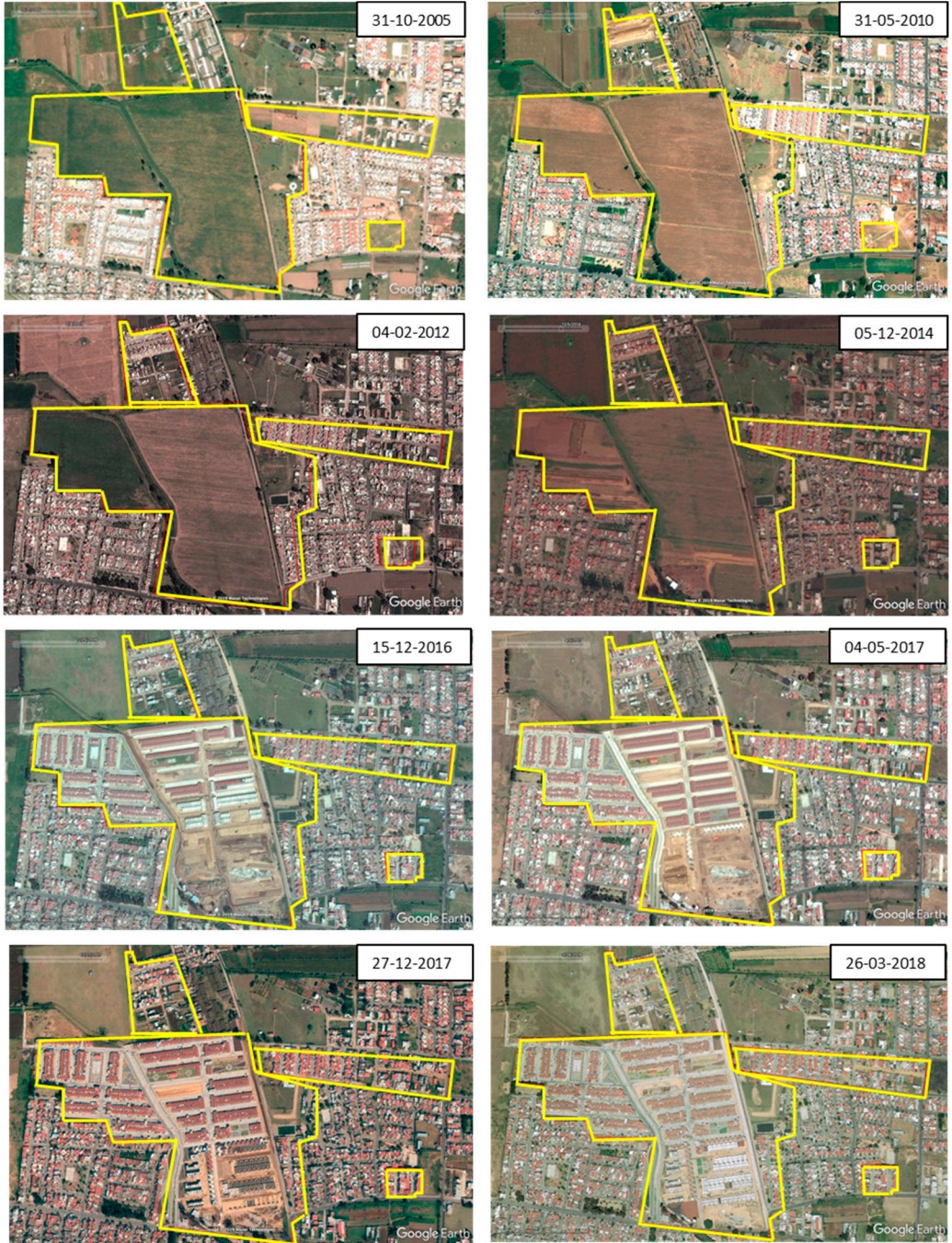

**Figure 12.** The urban expansion localized on the North-Western side of Ciùdad Guzman. The yellow lines indicate areas experiencing urban expansion in the 2005–2018 time period and, in particular after 2014. Time series of images with the Google Earth Time Slider tool provided by Google Earth Pro (v. 7.1.5.1557). The considered area refers to the black ellipses of Figure 4 and to the yellow ones in Figure 13.

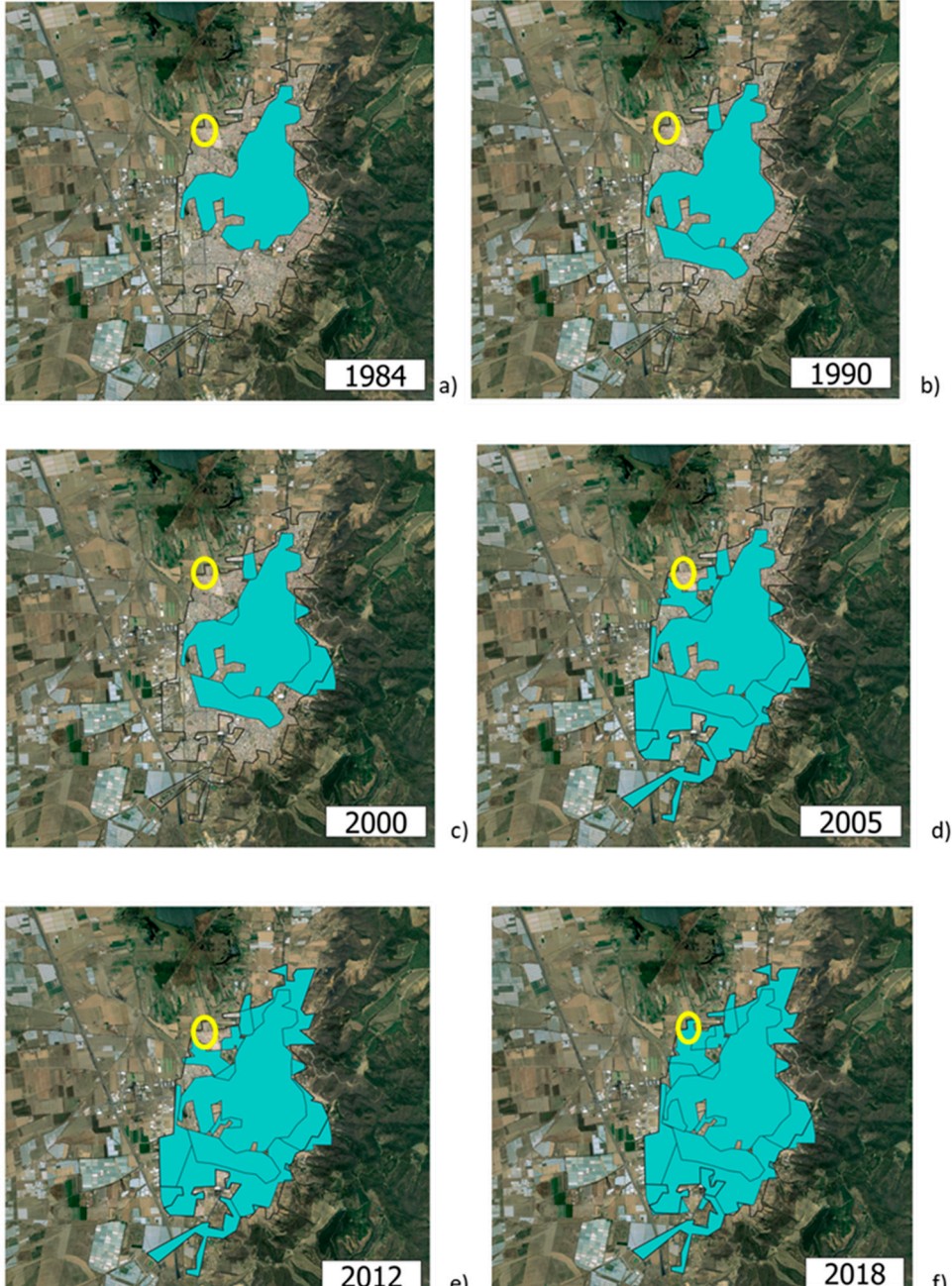

**Figure 13.** GC growth in the 1984–2018 period (**a**–**f**). The trends reported in Figure 14 relevant to the urban sprawl experienced by CG are relative to the surfaces evaluated from the polygons reported here. Time series of images with the Google Earth Time Slider tool (Google Earth Pro (v. 7.1.5.1557). Yellow ellipses identify the analysis in this section and reported in Figure 12.

Therefore, the urban sprawl can be associated with the subsidence that is still occurring in CG, but it is just a hypothesis because of the lack of a geotechnical model of the ground, meaning that we do not have a precise description of the soil response to the combination of stresses to which it has been subject and still is. We suggest that the origin of this phenomenon is related to a slow tectonic deformation combined with the lowering of the groundwater level, caused by an over-exploitation of the aquifer (likely due to the increase in the population) in the valley of the municipality of Zapotlán, characterized, at least for the most superficial layer, by granules with a low cohesion.

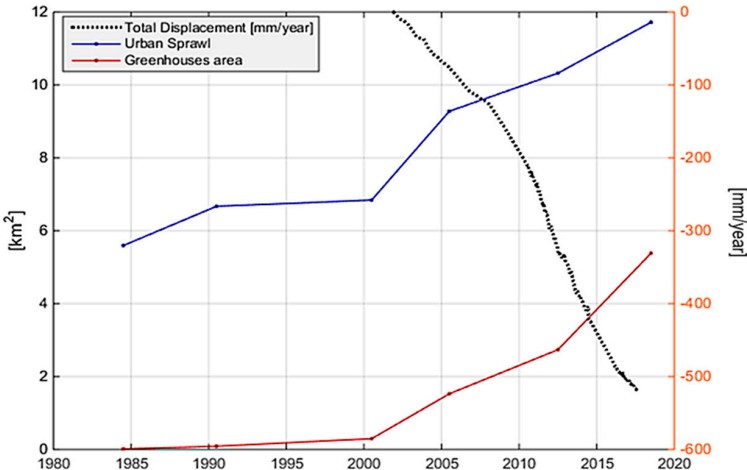

**Figure 14.** Trend of the urban sprawl in the municipality of Zapotlàn El Grande (blue dotted line) and of the growth of the greenhouse structures in square kilometers (red dotted line) compared with the deformation retrieved from the complete series of SAR data in the 2003–2018 period at site C (located in the northwestern part of CG, the one characterized by the highest deformation rates).

As far as the geological factors controlling the phenomenon, we can observe that the orientation of the fractures is parallel to the faults outcropping on the ridges bounding CG. Moreover, observing the S1 deformation map and the fault traced in the 2015 development plan [35] (Figure 15), it is clear that the fractures and the belt relative to the steep change in the deformation rate coincide with such structures. Also, in the hazard map reported in [36], the points relative to the area subject to geological hazards are located in correspondence of the steep deformation rate change retrieved by InSAR. This seems to confirm, together with the availability of InSAR data to retrieve subsidence phenomenon, especially in urban areas, the hypothesis that the observed fracture is the projection on the surface of a buried fault.

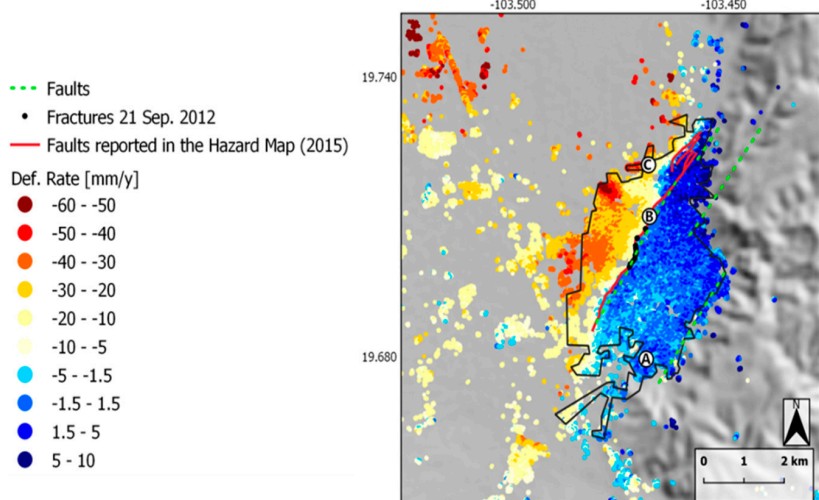

**Figure 15.** Overlay between S1 velocity map (ascending, values are expressed in [mm/year]) the fault mapped by the 2015 Development Plan (Government of the Municipality of Zapotlan el Grande [35]). The black lines are the faults traced in the framework of the "development map of the sub-district affected by the fault".

All these considerations allow stating that the increased subsidence rate in the 2016–2018 period shown by S1 data can be very likely ascribed to the over-exploitation of aquifers and to the following reduction of the underground water availability. The urbanization process, together with the changes

in the use of the soil predicted for the next 10 years, makes necessary a continuous monitoring of the ground deformation phenomenon.

## 6. Conclusions

Ground subsidence in the Northern Colima Graben and fissuring phenomena in Ciudad Guzmán were analyzed and updated in late 2018, together with the geological background of the city, by means of S1 SAR data. The displacement time series and the corresponding deformation rate fields were computed by two independent multi-temporal techniques, SBAS and PS, in order to cross-validate the obtained results. The deformation results estimated by the SBAS technique were integrated in a extended time series, together with ENVISAT and COSMO-SkyMed ascending outcomes, by re-projecting them onto the $LOS_{CSK}$, for consistency purpose. In addition, the vertical and the horizontal velocity fields were estimated from S1 ascending and descending data, confirming that the deformation continues to be mainly vertical, with respect to the lateral superficial motion component. The assumption that ground fissuring opened on 21 September 2012 because of an overexploitation of aquifers along the projection of a buried fault plane, is confirmed by the development plan of CG and by the natural hazard map published by the municipality of Zapotlán [36,37]. It is also highlighted by the strong discontinuity visible in the deformation field retrieved from InSAR measurements.

Finally, the retrieved results reflect the validity of InSAR displacement monitoring, but also the need of an accurate urban planning able to regulate land use, water withdrawal and agricultural activities. The need of a detailed environmental planning is fundamental in areas such complex in geological terms, as the CG is. Further work is foreseen in order to update the geological model in [13] with the deformation time-series obtained from the processing of the S1 stack here presented.

**Author Contributions:** Conceptualization, F.M., C.B. and C.A.B.; Data curation, F.M.; Methodology, C.A.B.; Project administration, C.B.; Software, F.M.; Supervision, C.B., C.A.B. and L.P.; Validation, F.M.; Writing—original draft, F.M.; Writing—review & editing, C.B., C.A.B. and C.T.

**Funding:** This research received no external funding.

**Conflicts of Interest:** The authors declare no conflict of interest.

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
