# Peer review of "Ground Deformations Controlled by Hidden Faults: Multi-Frequency and Multitemporal InSAR Techniques for Urban Hazard Monitoring"

_remotesensing, doi:10.3390/rs11192246_

Round 1
Reviewer 1 Report
Please see the attached PDF file

Reviewer 2 Report
The main objectives of the presented research were the subsidence in the densely build city in Mexico. Research area was chosen because of the geology influenced by the volcanic activity in this area. The earth fissures caused by the earthquake were observed with the use of the InSAR multitemporal and multispectral methods. Thanks a ENVISAT, COSMO-SkyMed and Sentinel 1 images the large span of movements from 2002 until 2018 was discussed. The correlation between the shapes of subsidence areas observing by the InSAR with faults position was the research niche of the manuscript. The research was discussed mainly clearly and concise. The abstract shows well the main objectives. The title of the manuscript seems to be a little bit confusing. The hazard identifying and management based on the monitoring results will have been expected according to the title. In the manuscript we have monitoring as well as the results – hidden faults identifying, but we do not have any chapter about the buildings, or infrastructure damages, nor damage assessment. Some of scientific approaches were devoted to the hazard research on the subsidence in the urban areas. It should be emphasized, that some of them try to assess the hazard or risk on buildings and infrastructure. I believe, it would improve the quality of the manuscript if the Authors will show those experiences in the introduction and will discuss the hazard in the discussion chapter. The second possibility would be changing the title of the manuscript. The research discussed in the manuscript is worth of publishing, the methodology was well described. The discussion of results clearly bases on the research. It is impressing that the long terms observations were possible thanks a sophisticated InSAR (SBAS and PS) methodology. Some of particular remarks could be consider by Authors. My particular remarks to the research were: 1) row 31: I doubt that the ground subsidence is a pure geological phenomenon. In some cases – yes (faults, natural compaction), but in many cases, it is rather an anthropological phenomenon caused by man activity in the rock-mass (mining, water withdrawal, oil- and gas exploitation). Is not it ? 2) rows 42, 230: not correctly linked references 3) Fig 9: the colors for the city development (1984 and 2018) are similar, which is difficult to distinguish. I believe that the research presented in the manuscript as a case study can be publish after the minor revision.
Round 2
Reviewer 1 Report
Lines 201-203:‘This is particularly true in the area where very fast subsidence is occurring: over these region the PS method, which does not unwrap the interferograms, is unable to “follow” such trend, thus it loses the points.’
I agree most of what the authors stated, but I could agree with the above one. The PS points were selected before phase unwrapping and won’t be discarded during phase unwrapping procedure. The possible problem of phase unwrapping for the fast-subsiding area could be phase unwrapping errors, but the PS points won’t disappear. You may explain this by the large temporal baselines in the previous sentences, but the area should not decorrelate that thoroughly if the baseline is the only reason. I will still advise the authors pay attention to the land cover of this region to see whether it’s vegetation-covered area to better explain the reason of disagreement.
Figure 4
You used (a) and (b) in the caption, but I did not see them on the figures.
